# Differential Effects of Lipid Bilayers on αPSM Peptide Functional Amyloid Formation

**DOI:** 10.3390/ijms25010102

**Published:** 2023-12-20

**Authors:** Kamilla Kristoffersen, Kasper Holst Hansen, Maria Andreasen

**Affiliations:** Department of Biomedicine, Aarhus University, Willhelm Meyer’s Allé 3, 8000 Aarhus, Denmark

**Keywords:** phenol-soluble modulins, functional amyloids, protein aggregation, protein–lipid interactions, lipid vesicles

## Abstract

Phenol-soluble modulins (PSMs) are key virulence factors of *S. aureus*, and they comprise the structural scaffold of biofilm as they self-assemble into functional amyloids. They have been shown to interact with cell membranes as they display toxicity towards human cells through cell lysis, with αPSM3 being the most cytotoxic. In addition to causing cell lysis in mammalian cells, PSMs have also been shown to interact with bacterial cell membranes through antimicrobial effects. Here, we present a study on the effects of lipid bilayers on the aggregation mechanism of αPSM using chemical kinetics to study the effects of lipid vesicles on the aggregation kinetics and using circular dichroism (CD) spectroscopy, Fourier-transform infrared (FTIR) spectroscopy and transmission electron microscopy (TEM) to investigate the corresponding secondary structure of the aggregates. We found that the effects of lipid bilayers on αPSM aggregation were not homogeneous between lipid type and αPSM peptides, although none of the lipids caused changes in the dominating aggregation mechanism. In the case of αPSM3, all types of lipids slowed down aggregation to a varying degree, with 1,2-dioleoyl-sn-glycero-3-phosphocholine (DOPC) having the most pronounced effect. For αPSM1, lipids had opposite effects, where DOPC decelerated aggregation and lipopolysaccharide (LPS) accelerated the aggregation, while 1,2-dioleoyl-sn-glycero-3-phospho-rac-(1-glycerol) (DOPG) had no effect. For αPSM4, both DOPG and LPS accelerated the aggregation, but only at high concentration, while DOPC showed no effect. None of the lipids was capable of inducing aggregation of αPSM2. Our data reveal a complex interaction pattern between PSMs peptides and lipid bilayers that causes changes in the aggregation kinetics by affecting different kinetic parameters along with only subtle changes in morphology.

## 1. Introduction

Functional amyloids are found in a wide variety of organisms in nature, where they provide a variety of functionalities ranging from storage of peptide hormones [1] to melanin granula formation [2] to virulence during infection [3] to forming scaffolds during biofilm formation [4,5]. Functional bacterial amyloids are secreted from bacteria as components of the biofilm, where they normally assemble into β-sheet-rich fibrils [6], providing stability to the biofilm [7]. In *S. aureus*, the functional amyloids are formed from small amphipathic peptides called phenol-soluble modulins (PSMs) [8]. Among other things, PSMs act as a structural scaffold in the biofilm matrix, increasing antibiotic resistance and biofilm integrity [9,10]. *S. aureus* produces seven PSMs called αPSM1–4, βPSM1–2 and δ-toxin [8]. The four α-group peptides consist of ~20 amino acids, and the two β-group peptides consist of ~40 amino acids [8]. αPSM1 and αPSM4 form prototypical amyloid fibrils which exhibit the typical cross-β structural signature, although with a great deal of structural polymorphism [11]. On the other hand, αPSM3 fibrils display a unique cross-α structure [12,13,14]. This structure enhances αPSM3 toxicity against human cells [15].

In addition to biofilm formation, the monomeric soluble PSM peptides have multiple functions including proinflammatory activity, cytotoxic activity by lysis of erythrocytes and neutrophils and antimicrobial activity [8,16,17,18,19,20]. In addition to in functional amyloid formation, αPSMs are also abundant on the cell surface, and expression of αPSMs on the cell surface of *S. aureus* promotes colony spreading [21]. Furthermore, community-associated methicillin-resistant *S. aureus* (CA-MRSA) has an extraordinary capacity to avoid being killed by neutrophils of the human innate immune system, even after phagocytosis by the neutrophils [22], and, additionally, PSMs contribute to phagosomal escape and intracellular lysis of the neutrophils [17,23]. The capacity to lyse human neutrophils is almost entirely dependent on the expression of PSMα genes [8]. The cytolytic activity of the PSM peptides differs, with αPSM3 displaying high cytolytic activity [24] and αPSM1 and αPSM2 only showing moderate activity and αPSM4 showing none [12,25]. Fibrillation, along with positive charges, plays a central role in αPSM3 cytolytic activity [14]. Unprocessed PSM peptides do not elicit antimicrobial activity in CA-MRSA [19]. Research shows that αPSM3 derivatives without large hydrophobic side chains elicit moderate-to-strong antimicrobial activity [26].

The complexity of functional amyloid formation in bacterial biofilms is increased by interactions between amyloid proteins and other components such as lipids, other proteins, nucleic acids and metal ions present in the biofilm [27]. Lipoproteins in human serum have the ability to inhibit PSM-induced lysis of neutrophils, with high-density lipoprotein (HDL) being the most potent one [28,29]. Cholesterol in the membrane (between 10 and 30 mol%) increases the lytic activity of αPSM1–4, and the ability of PSMs to form an α-helical structure is important for lysing cholesterol containing lipid vesicles [16]. Lipids have also been shown to affect other functional amyloid systems. The aggregation of curli-forming proteins CsgA and CsgB in *E. coli* is highly accelerated by the presence of lipopolysaccharide (LPS) [30], and the aggregation of FapC from *Pseudomonas* is also accelerated in terms of both increased growth rate and decreased lag phase in the presence of LPS [31]. The self-assembly of other amyloid proteins associated with pathology is also affected by the presence of various lipids. α-synuclein, involved in Parkinson’s disease pathology, has been shown to bind to lipid model membranes, resulting in a significant increase in aggregation rate, specifically the primary nucleation rate [32], and further investigation has shown that α-synuclein transitions from coil to α-helix when in contact with lipid bilayers [33]. Thus, lipid interactions with amyloid-forming peptides or proteins have been investigated multiple times in different settings, showing a connection between the two.

This notion makes it interesting to further investigate the interaction between lipid bilayers and PSM peptides since αPSM3 peptides have been shown to aggregate on the cell membranes, leading to membrane rupture and cell death [14]. Here, we investigated how different lipids could impact the aggregation of all the αPSM peptides. Since PSMs have antimicrobial activity [19], we additionally examined whether components in bacterial cell membranes affected the aggregation of PSMs.

## 2. Results

### 2.1. Chemical Kinetics Reveal Variation in Aggregation Rates for αPSM Peptides in the Presence of Different Lipids

To investigate the effect of lipids on the aggregation of αPSM peptides, we incubated peptides individually at fixed monomeric concentrations with varying concentrations of lipids. Aggregation kinetics were monitored using the amyloid-binding dye thioflavin T (ThT) [34]. We used lipid vesicles containing different lipids, namely, the zwitterionic 1,2-dioleoyl-sn-glycero-3-phosphocholine (DOPC) and the negatively charged 1,2-dioleoyl-sn-glycero-3-phospho-rac-(1-glycerol) (DOPG), and lipid vesicles mimicking bacterial cell membranes consisting of lipopolysaccharide (LPS) from Gram-negative bacteria and L-α-phosphatidylethanolamine (PE) with various fatty acid chains extracted from *E. coli*. All lipids were processed to mimic the lipid bilayer structure of cell membranes by forming unilamellar vesicles of uniform size. Peptide concentrations were chosen in accordance with previous PSM studies on aggregation in the absence of lipids [35].

For αPSM1, αPSM3 and αPSM4, reproducible aggregation curves were observed in the presence of DOPC, DOPG and LPS lipids (Figure 1). The presence of the lipid vesicles did not affect the ThT signal of αPSM peptide aggregates (Appendix A). A linear scaling of the end ThT fluorescence signal with respect to the monomer peptide concentration was seen for αPSM1, αPSM3 and αPSM4 (Appendix A). Kinetic data are thus presented as normalized ThT fluorescence curves with the *y*-axis converted to relative aggregate concentration. In the case of αPSM2, no aggregation occurred with any of the lipids (Appendix A). A summation of lipid effect on αPSMs is shown in Table 1.

The aggregation kinetics for αPSM1 did not significantly change when DOPG lipid vesicles were added (Figure 1b). However, changes in aggregation kinetics occurred when either DOPC or LPS vesicles were added to αPSM1. The aggregation rate decreased with higher DOPC concentrations, increasing the lag phase from ~25 h to ~50 h in a dose-dependent manner (Figure 1a). Interestingly, LPS vesicles had the opposite effect on the aggregation kinetics of αPSM1, and the aggregation was hence accelerated in a dose-dependent manner. At higher LPS concentrations (100–150 µg/mL), the lag phase was only ~2 h; therefore, the plateau was reached within ~25 h compared to ~40 h for the αPSM1 control sample in the absence of lipid vesicles (Figure 1c). The effect of lipids on the aggregation half-time can be seen in Appendix A.

The aggregation kinetics of αPSM3 were highly affected by the addition of lipid vesicles. Higher concentrations of all three lipids (DOPC, DOPG and LPS) all decelerated the aggregation kinetics of αPSM3, with DOPC lipid vesicles having the most profound effect. At higher concentrations of DOPC lipid vesicles, the lag phase was increased from ~10 min to ~250 min (Figure 1d). In the absence of lipid, the plateau phase was reached after ~50 min. This was increased to ~80 min with high concentrations of DOPG (Figure 1e), ~100 min with high concentrations of LPS (Figure 1f) and, with high concentrations of DOPC, the plateau phase was not reached even after ~320 min (Figure 1d). The deceleration of the aggregation kinetics of αPSM3 by addition of lipids was further confirmed by the increase in aggregation half-time as a function of lipid concentration (Appendix A), further confirming that DOPC lipid vesicles have the biggest effect on the aggregation kinetics of αPSM3.

In the presence of DOPC lipid vesicles, the aggregation kinetics for αPSM4 did not show any significant changes (Figure 1g). This behavior was also seen for LPS and DOPG lipid vesicles at low concentration, with no significant changes seen in the aggregation kinetics of αPSM4. However, at high concentrations of DOPG (100–150 µg/mL), the aggregation kinetics of αPSM4 appeared to be almost seeded as the lag phase disappeared (Figure 1h). In the presence of high concentrations of LPS lipid vesicles (80–90 µg/mL), the lag phase was still present but significantly shorter at 0–1 h compared to ~15 h for the control αPSM4 sample in the absence of lipids. The highest concentration of LPS lipids also caused the aggregation kinetics to appear to be seeded (Figure 1i).

In addition to DOPC, DOPG and LPS lipid vesicles kinetic experiments were also conducted with vesicles of PE lipids extracted from *E. coli* membranes to test whether lipids found in the outer leaflet of bacterial cell membranes could influence the aggregation of αPSM peptides. PE lipid vesicles had no effect on the aggregation of αPSM4, while aggregation of αPSM3 was completely inhibited by the presence of PE lipid vesicles. αPSM1 aggregation kinetics were slower in the presence of PE lipid vesicles; however, the aggregation kinetics in the presence of PE became un-reproducible, and single curves in a triplicate no longer followed the same trend. For these reasons, the aggregation in the presence of PE lipid vesicles was not further examined (Appendix A).

The differences in the lag phases between control samples of the same peptides were ascribed to batch-to-batch variations from the manufacturer, a phenomenon which has been observed before for synthetic peptides [36].

### 2.2. Fitting of ThT Curves Using Amylofit Reveals Different Mechanisms of Fibril Kinetics

To establish how the different lipids affected the molecular steps of aggregation in αPSM peptides, we used chemical kinetics to analyze the aggregation [37]. Kinetic models of protein aggregation have previously been applied for different model systems of protein aggregation [35,38,39,40]. Through the analysis of rates and reaction orders of underlying molecular events, the kinetic analysis of aggregation data renders it possible to determine the dominating molecular mechanism by which new aggregates are formed. We have previously shown that all αPSM peptides aggregate through a secondary-nucleation-dominated mechanism [35]. Kinetic parameters from this previous analysis of the aggregation of αPSMs in the absence of lipids were used as fixed global parameters where only one compound rate constant was individually fitted to the lipid concentration, either *k_n_k*_+_ or *k*_+_*k*_2_, with *k_n_* being the rate constant of primary nucleation, *k*_+_ being the rate constant of elongation and *k*_2_ being the rate constant of secondary nucleation. This approach has previously been used to establish how inhibitors influence specific microscopic steps during aggregation of other amyloidogenic proteins [41,42,43]. In some cases, a global fit for one of the compound rates, in addition to the compound rates allowed to vary with lipid concentration, was used to get a decent fit. A similar approach has previously been used to investigate the effect of heparin on the aggregation of PSM peptides [44]. The presence of lipids did not alter the dominating aggregation mechanism for any of the peptides, and all αPSM peptides were successfully fitted to a secondary-nucleation-dominated aggregation mechanism. The addition of heparin to βPSM2 has been observed to change the dominating aggregation mechanism from nucleation elongation dominated to secondary nucleation dominated. Fits to kinetic data are shown in Appendix A, and results from these fits are provided in Appendix A.

For αPSM1 and DOPC lipid vesicles, the best fits were obtained by allowing *k*_+_*k*_2_ to vary with every lipid concentration while *k*_+_*k_n_* was fitted as a global constant to all concentrations. The values of *n_c_* (reaction order of the primary nucleation) and *n*_2_ (reaction order of the secondary nucleation) were restricted to the control lipid-free values (Appendix A). Since both compound rates needed to differentiate from the control lipid-free kinetics, it is very likely that both *k*_+_ and *k*_2_ were affected by the presence of DOPC lipid vesicles. Interestingly, log(*k*_+_*k*_2_) values for the fit decreased in a linear fashion when plotted against the DOPC concentration (Figure 2a). When plotting the log of unfolding rates versus the concentration of denaturants or surfactant, similar linear relationships are found [45,46]. Since the concentration of the lipid was low (0.1 to 150 μg/mL), the interactions between the lipid and the PSM peptides were clearly strong in contrast to what is seen for denaturants which are used at high concentrations, indicating that the interactions occurring with proteins are driven by weak forces. The presence of DOPG lipid vesicles did not have a clear effect on the aggregation kinetics of αPSM1 (Figure 1b), which was further confirmed by the lack of a trend in the half-time plot (Appendix A). During the kinetic fitting analysis, the best fit was found by varying the *k*_+_*k*_2_ compound rate constant while globally fitting the *k_n_k*_+_ compound rate constant. The other kinetic parameters (*n_c_* and *n*_2_) were restricted to the lipid-free control values (Appendix A). This indicates that the rate constant which is affected by DOPG lipid vesicles is *k*_+_, although the changes are small, as also indicated by the small changes in *k*_+_*k*_2_ with lipid concentration (from 14.9 in the absence of lipids to 26.4 at the highest concentration of lipids, Appendix A). The obtained *k*_+_*k*_2_ compound rates can be fitted to a linear line in a log–log plot, although the R^2^ value (R^2^ = 0.69) further indicates a rather poor linear fit (Figure 2b). For αPSM1 and LPS, the best fit was obtained when *k*_+_*k*_2_ was varied with every lipid concentration and *k*_+_*k_n_*, *n_c_* and *n*_2_ were restricted to the lipid-free control values. This indicates that *k*_2_ is the rate constant affected the most since changes in *k*_+_ would also lead to changes in the second compound rate constant *k_n_k*_2_. A linear increase was seen when log(*k*_+_*k*_2_) was plotted against the LPS concentration, similarly to what was seen for DOPC and αPSM4 (Figure 2c,g).

For the kinetic data of αPSM3 and DOPC lipid vesicles, the best fit was found when *k*_+_*k*_n_ was allowed to vary with lipid concentration (Figure 2d). To obtain the best possible fit, *n_c_* and *n*_2_ values were also required to differ from the lipid-free control values. The values for *n_c_* and *n*_2_ used for the fit were found through trial by fitting of each parameter individually. Ultimately, *n_c_*, *n*_2_ and *k*_+_*k*_2_ were kept as global constants, meaning that *k*_n_ is the parameter most affected by the presence of DOPC lipid vesicles. This is also apparent from the kinetic curves, where the lag phase is clearly lengthened by additions of lipids; hence, DOPC vesicles clearly influence the primary nucleation of αPSM3. DOPG lipid vesicles also delay the aggregation kinetics of αPSM3, and the kinetic data were fitted with *k*_+_*k_n_* varying with lipid concentration. Additionally, the *k*_+_*k*_2_ compound rate constant also needed to be a global fit, along with *n_c_*, to obtain a good fit to the kinetic data. This indicates that both the primary nucleation and the elongation are altered in the presence of DOPG lipid vesicles. The decrease in the log(*k*_+_*k*_n_) followed a linear trend for αPSM3 in the presence of both DOPC and DOPG lipid vesicles (Figure 2d,e). For αPSM3 and DOPG kinetic data fitting, the *k*_+_*k_n_* varied with lipid concentration, leaving *k*_+_*k*_2_ as a global fitted parameter and *n_c_* and *n*_2_ as constant values. For αPSM3 and LPS, *k*_+_*k*_2_ was a fitted parameter with every concentration, leaving *k*_+_*k_n_* as a global fitted parameter and the lipid-free control parameters of *n_c_* and *n*_2_ as constant values. Since both compound rate constants were different from the lipid-free control values, it indicates that the parameter *k*_+_ is affected by the presence of LPS lipid vesicles. This is also apparent from the kinetic data where the slope of the elongation phase clearly changes with increasing lipid concentration. The decrease in *k*_+_*k*_2_ follows a linear pattern in a log–log plot when plotted against the lipid concentration (Figure 2f). Log–log patterns like this are similarly observed for ligand-binding systems, with a strong specific interaction between the ligand and the protein indicating a strong specific interaction between LPS lipid vesicles and αPSM3.

For all cases of kinetic data fitting to αPSM4 in the presence of lipids, bi-phasic behavior is seen with low concentration of lipids, giving either a small decrease in the kinetic compound rate constant or no changes at all. This is followed by an increase in the kinetic parameter at high concentrations of lipids regardless of the lipid. For αPSM4, the best fits for DOPC and DOPG were found when *k*_+_*k*_2_ varied with lipid concentration and the rest of the parameters were kept constant using the lipid-free control values (Appendix A). For αPSM4 and LPS, the best fit was found when *k*_+_*k_n_* was left to vary with lipid concentration and the rest were constant lipid-free control parameters (Appendix A). Log(*k*_+_*k*_2_) plotted against the concentrations of DOPC and DOPG reveals a bipartite graph initially with a decreasing (DOPC) or linear (DOPG) manner changing to an increasing linear pattern (Figure 2g,h). In the presence of LPS lipids, the log(*k_n_k*_+_) plot is bipartite, starting with a negative slope that turns into a positive slope at 60 μg/mL LPS (Figure 2i).

### 2.3. Lipids Have Modest Effects on the Secondary Structure of PSM Fibrils

To address whether adding lipids affected the secondary structure of the fibrillar aggregates of αPSM peptide, we used circular dichroism (CD) and Fourier-transform infrared (FTIR) spectroscopy. Individual CD spectra are shown in Figure 3a–i, and individual FTIR spectra are presented in Appendix A. Deconvolution of FTIR results is shown in Figure 4a–c.

αPSM1 has previously been shown to form β-sheet-rich fibrils [11,35], and smaller fragments of αPSM1 have shown polymorphism but with all fibril polymorphs being rich in β-sheet structures. For αPSM1, we also observed β-sheet-rich fibril structures based on the single minima at 219 nm in CD, indicative of a β-sheet structure, and, from the deconvolution of the FTIR data, more than 65% of the structure contribution was from β-sheets and β-turns. CD data and FTIR showed no changes in samples with αPSM1 and DOPG, which corresponds to the kinetic data where no significant changes occurred in the presence of DOPG lipid vesicles (Figure 3a and Figure 4a). FTIR of αPSM1 showed a slight change in the secondary structure of the fibrils when DOPC was added in a low concentration; however, no changes in structure were found at high concentrations of DOPC (Figure 4a). For αPSM1 and DOPC, the CD signal changed intensity, corresponding with the kinetic data which showed slower aggregation, which may have resulted in fewer fibrils being formed at the end stage of aggregation, resulting in a signal decrease in CD (Figure 3a). Alternatively, the decrease in signal intensity could also have been caused by the formation of fibrils with altered structures. LPS, on the other hand, accelerated the aggregation kinetics of αPSM1, and CD data showed a higher signal in samples with higher LPS concentration but with no changes to the wavelength of the minimum observed (Figure 3c). FTIR data showed an enrichment of the β-sheet structure of αPSM1 fibrils at low concentrations of LPS (Figure 4a).

The αPSM3 control sample aggregate showed a cross-α-helical structure consistent with previous studies [14,15]. For αPSM3, the CD spectra changed drastically for fibrils in the presence of all three lipids, and the changes were more pronounced with higher lipid concentration (Figure 3d,f). The signal intensity decreased, and the double minima signature of the signal indicates that the α-helical structure was less pronounced. Additionally, the minimum changed from 224 nm to 227.8 nm (Appendix A) for αPSM3 with a high concentration of DOPC (Figure 3d) and from 224 nm to 213.5 nm (Appendix A) for αPSM3 with a high concentration of LPS (Figure 3f). The minima changes correlated well with the FTIR data, which showed that the fibril of αPSM3 in the presence of all three lipids displays more β-sheet-rich structures (Figure 4b). For αPSM3 and DOPG, FTIR data revealed a complete change in structure as there was almost no α-helix left with the high concentration of DOPG (Figure 4b). For all three lipid samples, the signal intensity decreased, with higher lipid concentrations corresponding with the kinetic data where the aggregation was decelerated when adding lipids.

For αPSM4, the CD spectra showed no changes when DOPC was added (Figure 3g), which corresponds with the kinetic data where no significant changes in the kinetic curves were seen when DOPC lipid vesicles were added (Figure 1g). However, the FTIR data with DOPC showed a shift from β-sheets to β-turns at low concentrations of DOPC when compared to the lipid-free control sample (Figure 4c); this was also the case when LPS was added. For αPSM4, a change in secondary structure was seen when combined with either DOPG or LPS. The minimum was red shifted from 216 nm to 226.9 nm (Appendix A) for αPSM4 with added DOPG (Figure 3h) and to 222.5 nm when LPS was added (Figure 3i and Appendix A). The change in the minimum for αPSM4 with DOPG correlates well with the FTIR data, which showed a complete change to the α-helix only (Figure 4c). Interestingly, the CD signal intensity went down when the concentration of lipid went up, even though the kinetics showed a faster aggregation. This could indicate that, although the aggregation kinetics were accelerated, the amount of aggregated sample was less in the presence of high concentrations of DOPG.

### 2.4. Morphology Shows Fibril Formation When Adding Lipids to αPSM Peptides

The morphological features of aggregates from αPSM peptides with and without lipid were analyzed by transmission electron microscopy (TEM). Morphology of PSMs with low lipid concentration is shown in Appendix A. After incubation, αPSM1 formed long, entangled, amyloid-like fibrils, similar to what has been observed for αPSM1 previously [35] (Figure 5a). When 150 µg/mL of DOPC was added, we still saw entangled fibrils forming a network; however, the fibrils appeared shortened, and the network was less dense, consistent with the aggregation kinetics which are delayed in the presence of DOPC lipid vesicles. Lipid vesicles are also apparent in the images as a blur entangled with the fibrils (Figure 5b). In some areas where αPSM1 and DOPC were combined, large deposits of lipids were seen, and the morphology of the fibrils differed since the aggregates appeared decorated by amorphous aggregates (Appendix A). Despite no significant effect on the aggregation kinetics, adding 150 µg/mL of DOPG to αPSM1 resulted in a shortening of the fibrils along with the occurrence of less network structure, hence changing the fibril morphology despite similar aggregation kinetics. In some areas, lipids were observed occurring like a blur with aggregates close by (Figure 5c). Adding 100 µg/mL LPS to αPSM1 changed the fibril structure. The fibrils became shorter and thinner, and, in areas, the fibrils were combined with the LPS in larger structures where it was hard to establish the structure of individual fibrils (Figure 5d). αPSM3 generated shorter, ribbon-like and unbranched fibrils (Figure 5e), consistent with the morphology previously reported [35]. DOPC changed the number of fibrils seen in the TEM images. This correlates with the kinetic results, where DOPC was observed to have the largest impact by decelerating the aggregation to the largest extent. Furthermore, lipids were observed on the surface of the fibrils (Figure 5f). Although DOPG and LPS both decelerated the aggregation kinetics, the effect was less pronounced than in the presence of DOPC. This is also reflected in the TEM images, where significantly more aggregates were observed on the TEM grids for samples with DOPG and LPS compared to DOPC. In the presence of DOPG lipid vesicles, the αPSM3 aggregate morphology resembles the lipid-free control samples (Figure 5g). αPSM3 with 100 µg/mL LPS also resulted in fibrils resembling the control samples; however, LPS could be seen to associate with the fibrils (Figure 5h).

In the absence of lipid vesicles, αPSM4 formed fibrils with a morphology different from what has been reported for αPSM4 previously. The fibrils were shorter and thicker than in previous reports of αPSM4 (Figure 5i). When DOPC was added, the fibrils appeared thinner, and the network of aggregates was less dense (Figure 5j). DOPG lipid vesicles caused the αPSM4 fibrils to become even shorter, and there appeared to be lipid embedded in the fibril formation (Figure 5k). Despite the accelerated aggregation observed for high concentrations of LPS lipid vesicles, very few fibril structures were observed. The few fibrils present appeared to branch out from larger amorphous structures, and no fibril network was seen (Figure 5l).

## 3. Discussion

We studied the aggregation of αPSMs from *S. aureus* in the presence of single lipid vesicles, namely, DOPC, DOPG, lipopolysaccharide (LPS) from Gram-negative bacteria and PE from *E. coli*, using kinetic data and chemical kinetic analysis combined with secondary structure analysis using CD and FTIR and morphological analysis using TEM. Kinetic analysis revealed a varied effect of adding lipid to the PSM peptides; however, the effects were not homogenous between peptides and lipids. Analysis of the secondary structures revealed structural changes in agreement with the effects observed from the kinetic data.

In the *E. coli* curli system, LPS, found in the outer leaflet of Gram-negative bacterial cells [47], increases the rate of aggregation of CsgA and CsgB as a function of LPS concentrations [30]. We examined whether membrane lipids from bacteria could impact aggregation of αPSMs. The membrane lipid PE from *E. coli* did not accelerate aggregation of any of the αPSM peptides; in fact, PE blocked aggregation of αPSM3 all together (Appendix A). Hence, lipids from the membrane of bacteria are not able to aid in functional amyloid formation. This is the opposite effect to what is observed with respect to the LPS membrane component in the *E. coil* CsgA and CsgB system. The lack of effect from lipids from the bacterial membrane could possibly originate from the thick peptidoglycan layer found in Gram-positive bacteria [48]. This peptidoglycan layer could block the lipid surface from molecules found in the biofilm, and, hence, the bacteria might not benefit from accelerated aggregation of PSM peptides immediately adjacent to the cell membrane. Interestingly, LPS from the outer membrane of Gram-negative bacteria does influence aggregation when added to αPSM peptides. LPS accelerates the aggregation of both αPSM1 and αPSM4 but slows down aggregation of αPSM3. It is hard to determine whether the altered aggregation kinetics of αPSM3 are linked to the reported antimicrobial activity of αPSM3 [19] or whether other properties, e.g., structural properties such as the cross-α fibril structure, play a part [15].

It has previously been shown that the cytotoxicity effect of αPSM3 is promoted by aggregation and by introduction of positive charges in the amino acid sequence [14]. This could suggest that interaction between αPSM3 and negatively charged phospholipids would be favorable due to static attraction. However, the aggregation results in this study show that the charge of the lipids does not affect the fibrillation. All αPSM peptides have a net positive charge at neutral pH (isoelectric point (pI): αPSM1 = 9.7, αPSM2 = 10, αPSM3 = 9.5 and αPSM4 = 9.7), but all the peptides respond differently to the presence of the negatively charged DOPG lipid vesicles. No aggregation was induced for αPSM2, no changes in aggregation kinetics were seen for αPSM1, aggregation was decelerated for αPSM3 and aggregation was accelerated for αPSM4 in the presence of DOPG lipid vesicles (Figure 1). Furthermore, all other lipid tested here were zwitterionic, and they also elicited different responses from the peptides in terms of changes in the aggregation kinetics. Hence, our results show that the changes in aggregation kinetics in the presence of lipids are not governed by charges.

It has previously been showed that αPSM3 interacts with lipid bilayers in a species-specific manner, resulting in increased cytolytic activity for membranes mimicking mammalian cells [49]. It was furthermore reported that aggregation of αPSM3 was accelerated in the presence of lipid bilayers. Our current study shows aggregation of αPSM3 was slowed down with all lipids tested in a dose-dependent manner (Figure 1d–f). This is not in correlation with previous findings. However, different lipid compositions were used here; we used one single lipid for each system, compared to previous studies, where a mixture of lipids was used [49]. These differences in lipid composition could have led to the different effects observed here and in previous studies. Furthermore, in the previous studies, cholesterol was included in the lipid mixtures, and cholesterol clearly affects the PSM–lipid interaction [16]. Another explanation for the slower aggregation kinetics in our study could be the peptide–lipid interaction on the surface of the lipid vesicles leading to sequestration of peptide molecules to the lipid surface. This would lower the effective peptide concentration in solution.

The presence of the lipids in the current study did not induce aggregation of αPSM2. In normal in vitro conditions, αPSM2 does not aggregate by itself in aqueous solutions [35], and previous studies with heparin also failed in inducing aggregation of αPSM2 [44]. For αPSM3, aggregation promoted the cytolytic activity [12]; however, since αPSM2 has reported cytolytic activity but aggregation is not induced in the presence of lipid bilayers, the cytolytic activity of αPSM2 seems to be independent from aggregation. Interestingly, this seems to be the opposite effect for αPSM4. No cytolytic activity is reported for αPSM4, but we observed an acceleration of aggregation in the presence of high concentrations of DOPC and LPS lipids. Hence, no direct line can be drawn between the ability to lyse mammalian cells and aggregation of αPSM peptides.

## 4. Materials and Methods

### 4.1. Peptides and Reagents

N-terminally formylated peptides (>95% purity) were purchased from Genscript Biotech, Rijswijk, The Netherlands. Hexafluoroisopropanol (HFIP), thioflavin T (ThT), lipopolysaccharide (LPS) from Gram-negative bacteria and trifluoroacetic acid (TFA) were purchased from Sigma Aldrich, St. Louis, MO, USA. Dimethyl sulfoxide (DMSO) was purchased from Merck, Darmstadt, Germany. 1,2-dioleoyl-sn-glycero-3-phosphocholine (DOPC), 1,2-dioleoyl-sn-glycero-3-phospho-rac-(1-glycerol) (DOPG), L-α-phosphatidylethanolamine (PE) extracted from *E. coli*, extrusion filters and filter supports were purchased from Avanti Polar Lipids, Birmingham, AL, USA.

### 4.2. Preparation of Synthetic Peptide

Each dry lyophilized PSM peptide stock was dissolved to a concentration of 10 mg/mL in a 1:1 mixture of hexafluoroisopropanol (HFIP) and trifluoroacetic acid (TFA) followed by a 5 × 20 s sonication with 30 s intervals using a probe sonicator (Biologics, Inc., Manassas, VA, USA) and incubation at room temperature for 1 h. Further, the stock was divided into aliquots, and the organic solvent was evaporated using speedvac (Scanvac, Labogene Aps, Allerød, Denmark) at 1000 rpm for 3 h at room temperature. Dried peptide stocks were stored at −80 °C prior to use.

### 4.3. Preparation of Lipid Vesicles

Dry LPS lipid was dissolved in MilliQ water to a concentration of 1 mg/mL. PE, DOPC and DOPG were dissolved in chloroform. Chloroform evaporated overnight in an exicator, and dry lipid was dissolved in MilliQ water. LPS, PE, DOPC and DOPG lipids were then frozen in liquid nitrogen and thawed in warm water 10 times. The extruder heating block (Avanti Polar Lipids, USA) was placed on a hot plate until temperature was at 37 °C. The sample was loaded into one of the gas-tight 1000 µL glass syringes (Avanti Polar Lipids, USA) and put on the heating block to equilibrate for approx. 10 min. The sample was then passed through the extruder filter with a pore size of 0.1 µm 15 times, and the resulting unilamellar vesicles were ready to use.

### 4.4. Preparation of Samples for Kinetic Experiments

ThT fluorescence was observed in clear-bottomed half-area 96-well black polystyrene microtiter plates (Corning, New York, NY, USA) with a non-binding surface. We used a Fluostar Omega (BMG Labtech, Orthenberg, Germany) plate reader in bottom-reading mode. Aliquots of purified PSMs were thawed and dissolved in dimethyl sulfoxide (DMSO) to a concentration of 2 mg/mL prior to use. Concentrated peptide aliquots were diluted in MilliQ water and passed through a 0.22 µm filter. Lipid vesicles were added to the sample in different concentrations. ThT was added to the protein solutions to a final concentration of 40 µM. Further, samples were loaded (100 µL) in a 96-well plate and sealed to prevent evaporation. For each peptide, different concentrations—0.25 mg/mL (αPSM2 (110 μM) and αPSM4 (115 μM)) and 0.5 mg/mL (αPSM1 (222 μM) and αPSM3 (192 μM))—were used for kinetic measurements. The concentrations were chosen based on previous experiments made in the absence of lipids [35]. These concentrations allowed for observations of both acceleration and deceleration of the aggregation kinetics. The ThT:PSM peptide ratio was hence 1:6, 1:3, 1:5 and 1:3 for αPSM1, αPSM2, αPSM3 and αPSM4, respectively. The plates were incubated at 37 °C under quiescent conditions, and ThT fluorescence was measured every 10 min (20 s for PSMα3) with an excitation filter of 450 nm and an emission filter of 482 nm. The ThT fluorescence was followed for three repeats of each type of lipid concentration.

### 4.5. Far-UV Circular Dichroism (CD) Spectroscopy

At the end of ThT kinetics experiments, individual triplicate samples fibrillated in the absence and presence of high and low concentrations of DOPC, DOPG and LPS lipids were pooled. To remove the DMSO, samples were pelleted by centrifugation at 13,000 rpm for 30 min. The pellet was resuspended in the same volume of MilliQ followed by bath sonication. Samples were loaded in a 1 mm Quartz cuvette (Hellma, Müllheim, Germany).

CD was performed on a Chirascan Plus spectrophotometer (Applied Photophysics, Surrey, UK) at room temperature. Spectra were recorded between 185 and 250 nm with 1 nm bandwidth, step size of 0.5 nm, using three measurement repeats.

CD was performed on a JASCO-810 Spectrometer (Jasco, Oklahoma City, OK, USA) at 25 °C, wavelength 190–250 nm, with a step size of 0.1 nm, 1 nm bandwidth, 5 measurement repeats. For each sample, correction for baseline contribution and the MilliQ signal were subtracted.

### 4.6. Fourier-Transform Infrared Spectroscopy

Fourier-transform infrared spectroscopy was recorded on a Tensor 27 FTIR instrument (Bruker Optics, Billerica, MA, USA) equipped with an attenuated total reflection accessory with a continuous flow of N_2_ gas. To remove DMSO from the solution, fibrillated samples were centrifuged (13,000 rpm for 30 min), supernatants discarded and the pellet resuspended in MilliQ water. Prior to measurement, 5 µL samples were dried under a steam of N_2_ gas, and 64 interferograms were accumulated with a spectral resolution of 2 cm^−1^ in the range of 1000–3998 cm^−1^. OPUS 5.5 software (Bruker Optics, Billeria, MA, USA) was used to process the data, which included baseline subtraction, atmospheric compensation and a second derivative analysis. For comparative studies, all absorbance spectra were normalized. Only the amide I band in the range of 1600–1700 cm^−1^ comprising information about the secondary structure was shown.

### 4.7. Transmission Electron Microscopy

Fibrillated samples with and without lipid were collected following the ThT fibrillation kinetics assay by combining the contents of two to three identical wells from the plate. A 5 μL amount of all peptides with and without lipids was directly placed on a carbon-coated formvar grid (EM Resolutions, Keele, UK), allowed to adhere for 2 min and washed with MilliQ water followed by negative staining with 0.2% uranyl acetate for 2 min. Further, the grids were washed twice with MilliQ water and blotted dry using filter paper. The samples were examined using Morgani 268 (FEI Philips Electron microscope, Hilllsboro, OR, USA) equipped with CCD digital camera with a resolution of 1376 × 1032 and operated at an accelerating voltage of 80 KV.

## Figures and Tables

**Figure 1 ijms-25-00102-f001:**
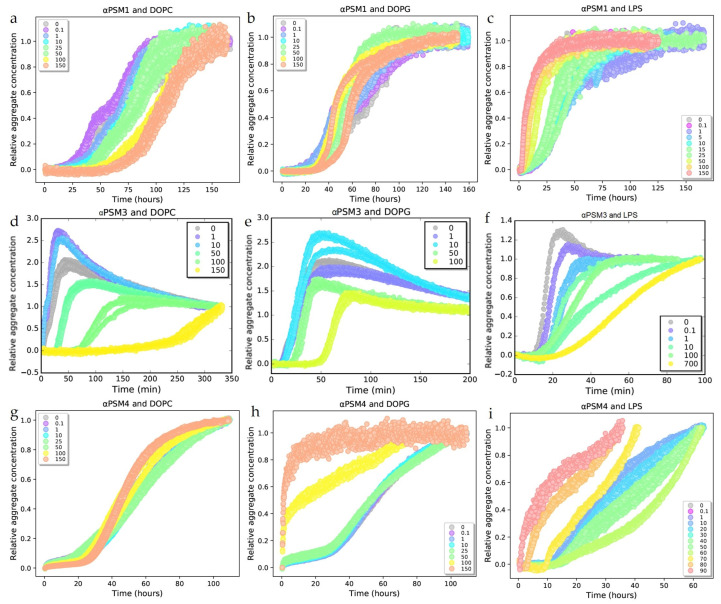
Experimental kinetic data for the aggregation of phenol-soluble modulin (PSM) peptides combined with lipids. Data are represented as normalized fluorescence curves. (**a**) Aggregation of αPSM1 (0.5 mg/mL) in the presence of DOPC lipid in different concentrations (0–150 µg/mL). (**b**) Aggregation of αPSM1 (0.5 mg/mL) in the presence of added DOPG lipid in different concentrations (0–150 µg/mL). (**c**) Aggregation of αPSM1 (0.5 mg/mL) in the presence of LPS lipid in different concentrations (0–150 µg/mL). (**d**) Aggregation of αPSM3 (0.5 mg/mL) in the presence of DOPC lipid in different concentrations (0–150 µg/mL). (**e**) Aggregation of αPSM3 (0.5 mg/mL) in the presence of DOPG lipid in different concentrations (0–100 µg/mL). (**f**) Aggregation of αPSM3 (0.5 mg/mL) in the presence of LPS lipid in different concentrations (0–700 µg/mL). (**g**) Aggregation of αPSM4 (0.25 mg/mL) in the presence of DOPC lipid in different concentrations (0–150 µg/mL). (**h**) Aggregation of αPSM4 (0.25 mg/mL) in the presence of DOPG lipid in different concentrations (0–150 µg/mL). (**i**) Aggregation of αPSM4 (0.25 mg/mL) in the presence of LPS lipid in different concentrations (0–90 µg/mL). All kinetic experiments were carried out in triplicates.

**Figure 2 ijms-25-00102-f002:**
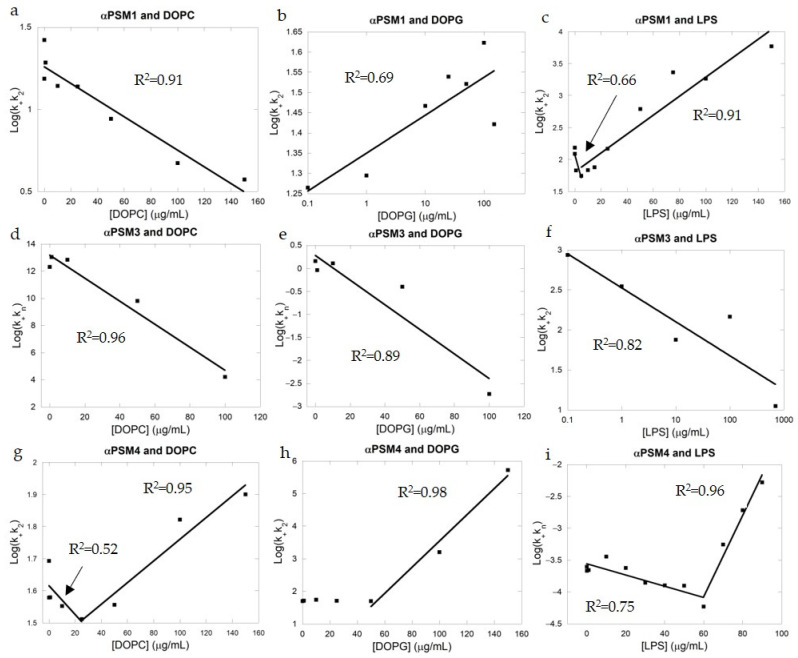
Plots of different composite rate constants obtained from fits to PSM aggregation data versus lipid concentration. (**a**) αPSM1 and DOPC. (**b**) αPSM1 and DOPG. (**c**) αPSM1 and LPS. (**d**) αPSM3 and DOPC. (**e**) αPSM3 and DOPG. (**f**) αPSM3 and LPS. (**g**) αPSM4 and DOPC. (**h**) αPSM4 and DOPG. (**i**) αPSM4 and LPS. The R^2^ of the linear fit to the data points (black squares) is given in each plot.

**Figure 3 ijms-25-00102-f003:**
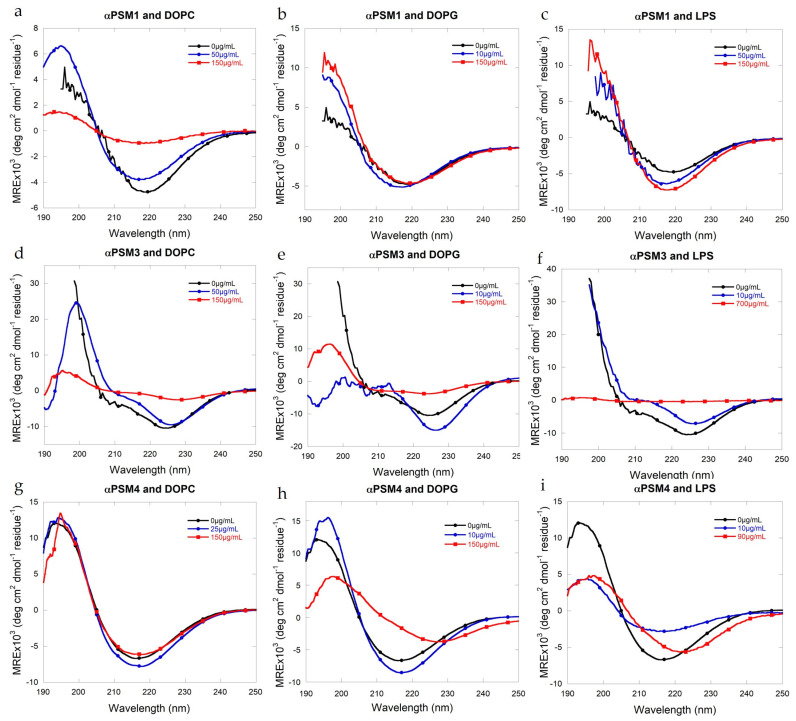
Structural comparison of fibrils formed by αPSM peptides when lipid vesicles are added. (**a**) Far-UV CD spectra for αPSM1 with added DOPC. (**b**) Far-UV CD spectra for αPSM1 with added DOPG. (**c**) Far-UV CD spectra for αPSM1 with added LPS. (**d**) Far-UV CD spectra for αPSM3 with added DOPC. (**e**) Far-UV CD spectra for αPSM3 with added DOPG. (**f**) Far-UV CD spectra for αPSM3 with added LPS. (**g**) Far-UV CD spectra for αPSM4 with added DOPC. (**h**) Far-UV CD spectra for αPSM1 with added DOPG. (**i**) Far-UV CD spectra for αPSM4 with added LPS.

**Figure 4 ijms-25-00102-f004:**
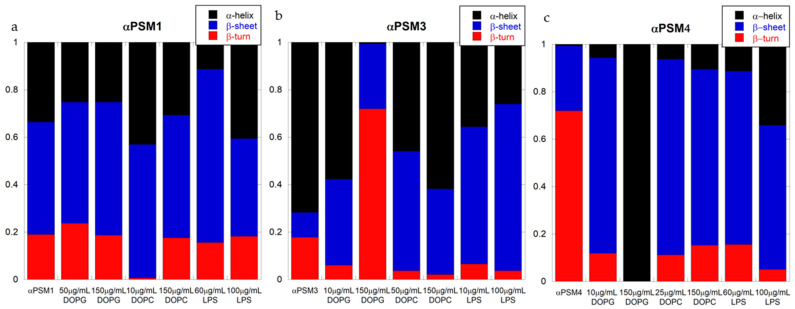
Deconvolution of the FTIR spectra of fibrils of the PSM peptides in the absence and presence of lipid vesicles. (**a**) PSMα1 with DOPG, DOPC and LPS at high and low concentrations. (**b**) PSMα3 with DOPG, DOPC and LPS at high and low concentrations. (**c**) PSMα4 with DOPG, DOPC and LPS at high and low concentrations.

**Figure 5 ijms-25-00102-f005:**
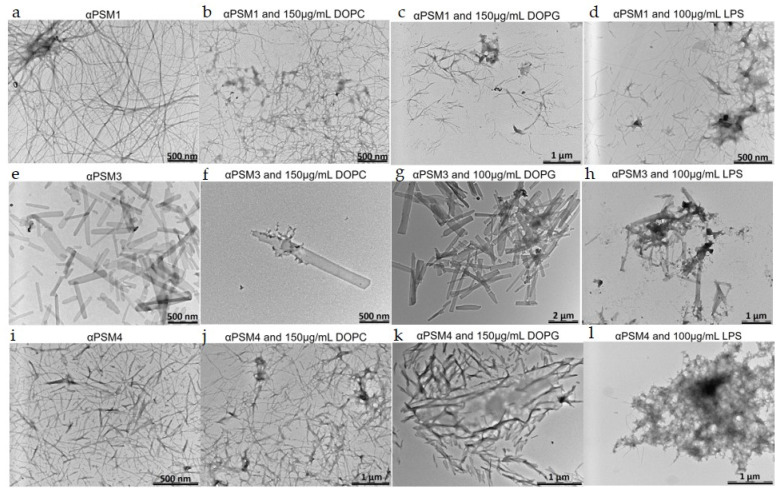
Morphology of aggregates of αPSM peptides with and without lipids. Transmission electron microscopic image of the end-state fibril of αPSM peptides in the absence and presence of DOPC, DOPG and LPS lipid vesicles. (**a**) αPSM1 fibrils. (**b**) αPSM1 with 150 μg/mL DOPC. (**c**) αPSM1 with 150 μg/mL DOPG. (**d**) αPSM1 with 100 μg/mL LPS. (**e**) αPSM3 fibrils. (**f**) αPSM3 with 150 μg/mL DOPC. (**g**) αPSM3 with 100 μg/mL DOPG. (**h**) αPSM3 with 100 μg/mL LPS. (**i**) αPSM4 fibrils. (**j**) αPSM4 with 150 μg/mL DOPC. (**k**) αPSM4 with 150 μg/mL DOPG. (**l**) αPSM4 with 100 μg/mL LPS.

**Table 1 ijms-25-00102-t001:** Summation of lipid vesicles’ effect on aggregation for αPSM peptides. ↑: Acceleration of aggregation kinetics, ↓: Deceleration of aggregation kinetics.

αPSM	DOPC	DOPG	LPS
αPSM1	↓	-	↑
αPSM3	↓	↓	↓
αPSM4	-	↑	↑

## Data Availability

The data presented in this study are openly available at FigShare: https://figshare.com/articles/dataset/Differential_effects_of_lipid_bilayers_on_PSM_peptide_functional_amyloid_formation/24494953, accessed on 3 November 2023.

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
