# Peer review of "Differential Effects of Lipid Bilayers on αPSM Peptide Functional Amyloid Formation"

_ijms, 2023, doi:10.3390/ijms25010102_

Round 1
Reviewer 1 Report
Comments and Suggestions for Authors
The authors used different experimental methods, such as CD,and FTIR and morphological analysis using TEM, to study the effects of various membranes on the aggregation of αPSMs. However, the molecular mechanism has not been clearly explained. The molecular mechanism of aggregation of αPSMs can be studied through molecular dynamics simulation methods, which should be added in the revised manuscript.
Reviewer 2 Report
Comments and Suggestions for Authors
The manuscript Differential effects of lipid bilayers on αPSM peptide functional amyloid formation studies the phenomenon of formation of amyloid structures in the presence of different types of components of lipid membranes (DOPC, DOPG, PE, the type of fatty acids is not indicated in this last case) and the predominant component of the bacterial wall in Gram-negative bacteria, such as LPS. The biophysical techniques used are varied: ThT fluorescence, circular dichroism, FTIR, TEM.
While the authors have done extensive work, I believe there is a great deal to review, making my decision a critical reworking on the authors' part. Based on the details I will give below, it will be at the editor's discretion whether the manuscript should be reworked and resubmitted, or if major changes will simply be requested, but in my opinion, the manuscript is not ready to be published in its original version.
My observations are the following:
Major concerns
1) The authors consider molecular effects when analyzing the formation or modulation of this phenomenon, but they start from erroneous molecular knowledge. They consider that PG is a lipid that characterizes the membranes of mammalian (eukaryotic) cells; However, this lipid is only synthesized in the mitochondria, which, according to the endosymbiotic theory, are of bacterial origin. In fact, only 1-2% of eukaryotic membranes are made up of PG (I recommend that you see the following review and the accompanying articles:
https://www.sciencedirect.com/topics/biochemistry-genetics-and-molecular-biology/phosphatidylglycerol).
What is even more serious is that the authors consider PG to be a positively charged lipid, when positively charged lipids do not exist in nature. In fact, since the glycerol head is neutral, the entire charge of PG comes from the phosphate group, which is negatively charged, making the authors' analysis completely invalid at this point.
2) The ThT fluorescence evaluation method is a known method for the evaluation of amyloid plaque formation, however, it is not completely specific. The authors detail the method in the corresponding section but do not explain why they consider this measure as “relative aggregate concentration” as they name it on the axis of the dependent variable. Whether there is any mathematical transformation or whether ThT fluorescence simply corresponds to this aggregation phenomenon is something that does not emerge from the manuscript.
In this sense, the assays lack controls, taking into account that the method is not specific, the presence of unilamellar liposomes could be modulating the fluorescence of ThT even in the absence of PSM, but this is not seen in the work.
On the other hand, in relation to this same method, the authors try to correlate the experimental data of the different methods, these being incongruous in several cases since they only consider that the change in fluorescence is about "non-formation" or the formation of amyloid to a lesser degree. However, effects such as the formation of different structures or super-aggregations, as would seem to be the case of PSM3 with DOPG, according to the TEM, are not considered by the authors. These new potential structures could manifest different affinities for ThT. However, at the fluorescence level, the authors always interpret it as a smaller amount of the macro-structure being formed.
3) Something that has caught my attention is the very different time range with which the kinetics of amyloid formation have been measured, being for almost all of them several days (~6 days), while for αPSM3 it has been only 6 hours. I think these measurements should be extended to the same range of time as the other two PSMs, since the same curves of αPSM3 (so different from the other two PSM cases) and mainly the yellow one, show that the process has not finished at that time, perhaps this is an initial event, especially considering that in the discussion, the authors claim that these particular αPSM3 results contradict those obtained by previous work by other authors. Furthermore, the presence of any of the membrane elements studied would seem to exert the same effect on the formation of αPSM3 amyloid.
4) In relation to the studies with PE, the authors do not reliably indicate what lipid it is, and this is not minor, since if it was DOPE, it has a tendency to form a hexagonal phase, unlike POPE or DPPE, which at working temperatures could be forming lamellae properly. The authors included the PE studies only in the supplementary material, since as stated in the manuscript (Page 4, lines 157-158) “For these reasons the aggregation in the presence of PE lipid vesicles was not further examined (Figure S1a-c).”. However, in the discussion conclusions are drawn about it. This is not appropriate for me.
On the other hand, they consider PE as a mimetic model of gram-positive prokaryotic cells, but it is far from being adequate, since PE is one of the main components of the internal membranes of eukaryotic cells, and the lipid mimetic models documented of bacteria to differentiate the membranes of gram-positive and negative bacteria, are different ratios of PG:PE.
Minor concerns
i) The caption of Figure 1 is inadequate, it mixes the presentation of the figures with the discussion about them, it should be adapted to what is considered a caption.
ii) The acronyms must be explicit the first time they appear since the works are not always read by specialists in the same techniques that are used.
iii) In the Materials and Methods section, when explaining the fluorescence method, the authors should indicate the probe:protein ratio in molar ratios, as well as indicate the reason why different concentrations are used for different PSMs evaluated.
For all the above, I consider that the changes that the authors should make to adequately adjust the manuscript to a publishable version are too many. Therefore, in my opinion, the manuscript should be rejected in its current version and resubmitted with the corresponding comments solved. I remain available for a new evaluation if the editor considers it so.

Round 2
Reviewer 1 Report
Comments and Suggestions for Authors
The paper can be accepted in present form.
Author Response
We sincerely thank the reviewer for the valuable input and suggestions put forward during the review of the manuscript
Reviewer 2 Report
Comments and Suggestions for Authors
I note that the authors have made the suggested changes. However, they have introduced a new error, which they do not transfer to the rest of the text, but which gives the idea that the authors do not really have a molecular management of membrane lipids.
Where says:
We used lipid vesicles containing different lipids namely the negatively charged 1,2-dioleoyl-sn-glycero-3-phosphocholine (DOPC) and the zwitterionic 1,2-di-oleoyl-sn-glycero-3-phospho-rac-(1-glycerol) (DOPG),
It should say:
We used lipid vesicles containing different lipids namely the zwitterionic 1,2-dioleoyl-sn-glycero-3-phosphocholine (DOPC) and the negatively charged 1,2-di-oleoyl-sn-glycero-3-phospho-rac-(1-glycerol) (DOPG),

Author Response
We thank the reviewer for pointing this mistake out to us. We apologize for the error. We have changed the text to reflect the true charge of the lipids and used the changes suggested by the reviewer